# Unexpected Value of Honey Color for Prediction of a Non-Enzymatic H_2_O_2_ Production and Honey Antibacterial Activity: A Perspective

**DOI:** 10.3390/metabo13040526

**Published:** 2023-04-06

**Authors:** Katrina Brudzynski

**Affiliations:** 1Department of Drug Discovery, Bee-Biomedicals Inc., St. Catharines, ON L2T 3T4, Canada; beebio@sympatico.ca; 2Department of Biological Sciences, Formerly, Brock University, St. Catharines, ON L2S 3A1, Canada

**Keywords:** honey color, pro-oxidative activity, H_2_O_2_ production, redox cycling, polyphenols, flavonoids, transition metals, Fe, Cu, radical scavenging, metal chelation, melanoidins, colloids

## Abstract

Hydrogen peroxide is the principal antibacterial compound of honey and its concentration determines honey bacteriostatic (MIC) and bactericidal (MBC) potencies. Levels of H_2_O_2_ produced are highly relevant to honey therapeutic potential, but they vary extensively among honey with reasons not immediately apparent. According to a traditional view, H_2_O_2_ is produced as a by-product of glucose oxidation by the honey bee enzyme, glucose oxidase; however, significant levels of H_2_O_2_ could be produced in a non-enzymatic way via polyphenol autooxidation. The aim of this study was to evaluate the potential for such an alternative pathway by re-examining evidence from many experimental and correlative studies in order to identify factors and compounds required for pro-oxidant activity. Unexpectedly, the color intensity was found to be the main indicator separating honey varieties based on the quantitative differences in the polyphenolic content, antioxidant activity and the content of transition metals, Fe, Cu and Mn, the main factors required for pro-oxidant effects. The color-impeding polyphenolics and their oxidation products (semiquinones and quinones) further contributed to color development through multiple chemical conjugations with proteins, phenolic oxidative polymerization, chelation or the reduction of metal ions. Moreover, quinones, as an intrinsic part of polyphenol redox activity, play an active role in the formation of higher-order structures, melanoidins and colloids in honey. The latter structures are also known to chelate metal ions, potentially contributing to H_2_O_2_ production. Thus, the color intensity appears as a major parameter that integrates polyphenol-dependent pro-oxidant reactions resulting in H_2_O_2_ generation.

## 1. Introduction

Hydrogen peroxide is the main antibacterial compound of honey. From the date of its discovery as “inhibine”, (a substance that inhibited bacterial growth) by Dold, Du, Dziao, 1937 [1] and its identification as hydrogen peroxide [2], multiple studies confirmed the validity of the causal association between the concentration of H_2_O_2_ and honey potency as an antibacterial agent [2,3,4,5,6,7,8,9,10,11,12].

Although honey possesses various secondary metabolites with antimicrobial activities originating from plants (flavonoids, phenolic acids, volatile compounds, pathogenesis-related proteins), microbes (antimicrobial peptides, biosurfactants, siderophores) and honeybees (defensins, hymenoptaecins, jelleins) [13], none of these compounds alone can kill or suppress the growth of a broad spectrum of microorganisms to a degree comparable with H_2_O_2_. Removal of H_2_O_2_ by pre-treatment of honey with catalase reduces honey antibacterial activity by 50–80%, reinforcing evidence of its major contribution to honey antibacterial activity [6,11,14]. 

The H_2_O_2_ production and accumulations showed high variability among different honeys with reasons not immediately apparent. The reported concentrations of H_2_O_2_ can vary up to 100 times, ranging from 0.04 to 4 mM, with an average of 2.5 mM [7,15,16]. Since H_2_O_2_ synthesis and its magnitude differ between honey varieties, this suggests that compounds involved in H_2_O_2_ accumulation and H_2_O_2_ degradation represent individual properties of honey deriving from honey botanical origin. (This review concerns honeys of European and North American origins and does not include Manuka honeys of New Zealand and Australia, derived from *Leptospermum scoparium,* which is apparently devoid of H_2_O_2_).

The main pathway of H_2_O_2_ production in honey is via enzymatic synthesis during glucose oxidation by bee glucose oxidase [2,13]. Additional sources of H_2_O_2_ production in honey may comprise glucose oxidases of microbial and plant origins [17,18].

However, another underexplored source of H_2_O_2_ synthesis in honey is its non-enzymatic synthesis via polyphenol autooxidation. The amount of H_2_O_2_ produced in this reaction depends on polyphenol composition and concentrations which vary among honeys of different botanical origins. The quantitative differences in polyphenols capable to autoxidize could explain the observed differences in the levels of H_2_O_2_ produced among honeys. The process of H_2_O_2_ generation by polyphenol autooxidation is observed in tea, coffee, red wine and fruit juices [19,20,21,22]. High polyphenolic content in honeys implies the possibility that a non-enzymatic pathway of H_2_O_2_ synthesis might also take place in this food. 

Polyphenol autooxidation is viewed as an undesirable outcome due to the generation of radical oxygen species with unpaired electrons (ROS), superoxide radicals, hydroxyl radicals and H_2_O_2_, causing oxidative stress in living cells. However, in honey, this pro-oxidant effect may have a profoundly positive role by evoking oxidative stress in bacterial cells, thereby increasing honey’s bactericidal potency. This notion comprises a strong rationale to evaluate the potential for the non-enzymatic production of H_2_O_2_, its mechanism, and participating components of the reactions and conditions. The specific aim was to select and define honey physicochemical parameters to best predict the non-enzymatic H_2_O_2_ production. This was accomplished by reviewing and summarizing experimental and correlative data previously collected on the relationship between the concentration of color-impeding polyphenols with honey color and antioxidant/prooxidant activity and the content of transition metals. 

## 2. Correlation between H_2_O_2_ Production and Color of Honey Variety as a First Indication of the Pro-Oxidant Activity

The production of H_2_O_2_ is not linear with honey dilutions and appears in a narrow range, usually between 30 and 50% *v/v* [2,9,16,23,24]. In some dark honeys, such as buckwheat, heather, and chestnut, the peak of H_2_O_2_ production occurred at higher honey dilutions, between 6.25 and 12.5% (*v/v*). Yet, some light honey varieties, such as sweet clover, acacia, rapeseed honey, multifloral honeys and honey blends showed a peak at the 25–50% range of dilution, indicating the contribution of pigmented compounds, such as polyphenols, flavonoids or carotenoids, to H_2_O_2_ production [9,16,24]. The relationship between H_2_O_2_ production and honey dilution assumes an inverted U-shaped curve [24].

More details on the relationship between the color of honey variety and H_2_O_2_ production were obtained from our correlative study. We have been able to demonstrate statistically that the production and accumulation of H_2_O_2_ differ significantly among three distinct color groups; dark-, medium- and light-colored honeys (Figure 1, Table 1). All honeys used were fresh, fluid and uncrystallized. Dark honeys produced significantly higher amounts of H_2_O_2_ than medium and light honeys (two-way ANOVA, F (2,36) = 308.60, *p* < 0.001, Figure 1). The content of H_2_O_2_ (measured by Amplex Red method) was dependent on honey dilution in a highly-significant way (ANOVA, F (3,36) = 115.46, *p* < 0.00), and its concentrations were the highest between four and eight times dilutions (25–12.5% (*v/v*)) in different color honeys. With further dilutions, the content of H_2_O_2_ rapidly decreased (Figure 1). This decrease occurred along a significantly different curve shape for dark- medium- and light-colored honeys as shown by a highly significant interaction between color and dilution (*p* < 0.001). Although the content of H_2_O_2_ in dark- and light-colored honey decreased from maximal to lowest values by a similar percentage (68–70%), the production/accumulation of H_2_O_2_ in dark- and medium-colored honeys first increased and then sharply decreased at 16–32 times dilutions (6.25–3.125% *v/v*) (blue and red bars in Figure 1). On the other hand, the light-colored honeys showed rather uniform and F (3,36) = 115.46, *p* < 0.001 almost linear decreases across all dilutions (green bars in Figure 1). 

The significant differences in the H_2_O_2_ contents between dark- medium-and light-color honeys revealed that compounds influencing honey color participate in the H_2_O_2_ production.

## 3. The Interrelationship between the Content of Pigmented Polyphenols, Honey Color, and H_2_O_2_ Production 

The color distinctions between dark and light honeys are attributed to the differences in the concentration of pigmented compounds such as flavonoids, their quinones, carotenoids and the colored Maillard Reaction Products (MRPs) that gives the yellow, orange, red and brown colors to dark honeys [25,26]. In addition, a significant aspect in the dark color development arises from the redox activity of polyphenols. The multiple color-changing reactions in which phenolic compounds participate include polyphenol oxidation to semiquinones and quinones that produce a range of colors from yellow to brown [27,28,29], quinone-protein complexation that produces dark reaction products, polyphenol oxidative polymerization that increases the conjugated double- bonds system contributing to browning [30], the formation of colored Maillard reaction products [31], complexation reactions with iron [27,32] and the formation of brown, high molecular weight melanoidins [33]. Therefore, it was reasonable to propose that polyphenol autoxidation is the reaction that could explain the relationship between honey color and H_2_O_2_ production. 

In the quest for finding support for this hypothesis, we reviewed a large number of multivariate analyses to uncover correlations between honey components and factors through which they can contribute to the formation of H_2_O_2_ via pro-oxidative activity. While there are an immense number of publications on the antioxidant activity of honey polyphenols, research on their pro-oxidant activity is scarce. There is a subtle balance between antioxidant and pro-oxidant activity that can be changed depending on the environmental conditions [34]. The antioxidant activity of polyphenols is transformed into pro-oxidant activity in the presence of transition metals or increased concentration of ROS or strong metal-chelating compounds (see Section 7). Therefore, the re-examination of the relationship between the polyphenol content, antioxidant/pro-oxidant activities and honey color might be relevant for non-enzymatic H_2_O_2_ production.

### 3.1. Honey Color Appeared as a Dominant Parameter in Honey Classifications in Multivariate Analysis

The antioxidant activity of honey polyphenols has been extensively studied, generating a huge number of papers, to name a few highly cited publications [35,36,37,38,39,40,41,42,43,44]. Various multivariate statistical methods were used for that purpose; Pearson correlation coefficient, principal component analysis (PCA), linear discriminant analysis (LDA), cluster analysis (CA), neural network (NN) and their combinations [45,46,47,48,49,50,51,52,53,54,55]. Because the chemometric approach was often employed for different purposes such as to differentiate honeys according to their botanical or geographical origin or to distinguish genuine honey from adulterated honey, the choice of the variables was broad and included, between others, honey color, the total phenolic content (measured by Folin–Ciocalteau method, FC), flavonoid content (measured by aluminum chloride method) and antioxidant activity (measured by DPPH (2,2-diphenyl-1-picryl-hydrazyl-hydrate), FRAP (Ferric Reducing Antioxidant Power) or ORAC (Oxygen radical absorbance capacity assay), electrical conductivity, ash content, pH, free acidity and glass transition temperature [26,51,53,54,55]. Despite different variables being tested in multivariate analyses, honey color has been repeatably found to be the most discriminant parameter to distinguish and separating honey varieties [26,51,53,54,55]. Table 2 provides a summary of collated data from over 50 multivariate analyses emphasizing very strong correlations (r > 0.83, *p* < 0.05) between honey color, the total phenolics content (TPC), total flavonoid content (TFC) and antioxidant activity measured by ABTS (2,2′-azinobis-(3-ethylbenzothiazoline-6-sulfonate), FRAP, DPPH or ORAC methods. 

In general, darker colors of honey had a higher content of polyphenols and higher antioxidant activity than light-colored honeys [36,37,38,39,40]. When honeys were arranged according to the increased color from the extra light to dark amber, they displayed a proportional increase in the total flavonoids and the total polyphenol content, indicating that polyphenolics contributed to both honey color and honey antioxidant activity [48,49]. The evidence that emerged from these correlations indicates that the above parameters are directly interdepended of each other.

### 3.2. Correlation between Honey Color, Polyphenol Content and H_2_O_2_ Production

The strong correlation between antioxidant activity, polyphenol content and honey color raised the question of whether there is a similar, strong interrelationship between the polyphenol content, honey color and H_2_O_2_ production in honeys? 

We have earlier indicated that honey color can differentiate honey groups based on the level of H_2_O_2_ production (Figure 1). Using the data of Table 1, the correlation between four parameters (e.g., honey color, polyphenolic content, antioxidant activity and H_2_O_2_ production) was tested with a bivariate Pearson correlation coefficient on honeys presented in Table 3. The correlation matrix showed again the highly significant correlation between honey color and the total phenolic content (r = 0.971, *p* < 0.0001), color and antioxidant activity (r = 0.8892, *p* < 0.0001), in addition to the known, strong correlation between the total phenolic content and antioxidant activity (r = 0.9794, *p* < 0.0001) (Table 3). On the other hand, the strength of the relationship between the color of honey and H_2_O_2_ content (r = 0.6512, *p* = 0.0218), H_2_O_2_ content and the total phenolic content (r = 0.6826, *p* = 0.0144) and H_2_O_2_ content and antioxidant activity (r = 0.6823, *p* = 0.0145) was of a moderate, positive significance. These preliminary data might suggest the existence of other, unidentified factor(s) affecting the relationship between these variables. Because the non-enzymatic production of H_2_O_2_ requires the presence of transition metal ions such as Fe^3+,^ Fe^2+^, Cu^2+^ and Cu^+^ and that metal ions and their reaction products strongly contribute to the honey color, it is plausible that metal ions are the missing factor.

## 4. Basic Concept of H_2_O_2_ Production via Polyphenols Autoxidation

To rationalize the hypothesis of the pro-oxidant activity of polyphenols as the alternative pathway of H_2_O_2_ formation in honey, it is necessary to describe in general terms, the mechanism of polyphenol autooxidation.

Literature data indicate that the pro-oxidant effect requires the presence of polyphenols containing structural elements that favor pro-oxidant activity, transition metal ions, specifically Fe^3+^ and Cu^2+^ and oxygen [56]. The structural elements necessary for both antioxidant and pro-oxidant activity of polyphenols are identified as (a) a catechol group on the B-ring (b) the 2, 3-double bond in conjugation with a carbonyl group at 4-position in the C-ring, and (c) the presence of hydroxyl groups at the 3 and 5 positions (Figure 2A) [57,58]. The propensity to pro-oxidation and the production of H_2_O_2_ are strongly influenced by multiple hydroxyl groups in the B-ring. Catechol (3′, 4′-dihydroxy) and pyrogallol (3′, 4′, 5′ -trihydroxy) structures, such as those in quercetin or myricetin, respectively, determine the maximal proton-coupled electron-donating effect (Figure 2B) [56,59].

Based on the literature data, the proposed mechanism of non-enzymatic H_2_O_2_ production by auto-oxidation of flavonoid quercetin is schematically presented in Figure 2. The autooxidation of quercetin results from the one-electron donation from quercetin to molecular oxygen via Fe^3+^ as a catalyst, giving rise to a semiquinone radical and superoxide anion radical (superoxide (O_2_^•−^). The formation of a semiquinone radical anion is a prerequisite for the generation of O_2_^•−^. The appearance of O_2_^•−^ is evidence of pro-oxidant activity. The reaction between molecular oxygen and the catechol group of quercetin is facilitated by transition metals, Fe^3+^ or Cu^2+^ that donate one electron directly to O_2_ [20,56]. In addition to Fe^3+^ or Cu^2+^, a superoxide radical (O_2_^•−^), or hydroxyl radical (^•^OH) can act as reducing agents, oxidizing flavonoid to semiquinone radical. The superoxide radical generated in this reaction reacts with other superoxide radicals and, in the presence of protons or water, H_2_O_2_ is formed through dismutation (one superoxide is reduced to hydrogen peroxide and the other is oxidized to dioxygen) [61,62]. 

By a virtue of redox cycling, the oxidized semiquinone radical can be restored to quinone by donating one electron to superoxide (O_2_^•−^) and generating H_2_O_2_ by dismutation. Redox cycling would start again and become established when the restored quinone becomes reduced again to semiquinone radicals by the reaction with a previously formed superoxide radical, which in turn, would generate another flux in H_2_O_2_. The rate of reaction of the semiquinone radical with O_2_ (in the presence or absence of transitional metal ions) determines the rate of the H_2_O_2_ formation (Figure 2). The generation of quinones and stable semiquinones radicals, the presence of transition metal and reducing agents (superoxide radical or hydroxyl radical) can accelerate the oxidation process by redox cycling producing substantial amounts of H_2_O_2_. 

## 5. Presence of Phenolic Acids and Flavonoids of Required Structures for Pro-Oxidant Activity in Honey 

To consider the possibility of H_2_O_2_ production in honey via polyphenol autoxidation requires the assurance that honeys, specifically dark honeys, contain polyphenols bearing the required structural elements to promote pro-oxidation and transition metal ions (Figure 2A). Polyphenol profiling of dark honeys using a high-resolution LC-MS (such as UPLC-PDA-MS/MS, UHPLC- LTQ OrbiTrap MS) [46,63,64] revealed that dark honeys, buckwheat, chestnut, heather and Agastache honey all possessed a significant concentration of p-hydroxybenzoic, chlorogenic, caffeic acids in addition to phenyllactic, p-coumaric, abscisic and ellagic acids [64,65,66,67,68]. The presence of a catechol group in caffeic and chlorogenic acids strongly influenced the antioxidant value of these compounds measured by DPPH, FC, FRAP, or ORAC. On the other hand, phenolic acids with only one hydroxyl group such as p-coumaric and hydroxybenzoic acid showed lower antioxidant values [59]. 

Similarly, among honey flavonoids, a predominant group in dark heather, buckwheat and chestnut honeys comprised the flavone chrysin (5,7-*dihydroxyflavone)*, flavanone pinocembrin and flavonols galangin and myricetin derivatives and glycosylated flavonols, quercetin rhamnoside, rutin and kaempferol rhamnoside [48,67,68,69]. Regarding their antioxidant/pro-oxidant activities, flavonols (galangin, myricetin, quercetin, kaempferol) were found the most active among flavonoids, while flavones (chrysin) and specifically flavanone (pinocembrin) was weak antioxidant due to a lack of catechol structure and C2–C3 double bond [59,70]. However, multivariate analyses showed several other flavonoids and phenolic acids with strong antioxidant/pro-oxidant activities that were shared among many honeys, in addition to the flavonoids serving as specific markers of some honey varieties [71]. Importantly, new research showed a cooperative action between flavonols (quercetin or rutin) and flavanones (naringenin and naringin) that increased the ability of flavanones to donate electrons and oxidize [72].

Thus, the synergistic action of polyphenolic acids and flavonoids rather than an individual compound could play role in the pro-oxidant activity and generation of H_2_O_2_ in honey. 

In honey, numerous quinone types can be formed at the same time having opposite anti-or pro-oxidant effects. The quinone concentrations and the concentrations of transitional metals would determine the pro-oxidant or antioxidant outcome [73].

## 6. Presence of Transition Metals

Redox cycling indicates that the antioxidant and pro-oxidant reactions of polyphenols are reversible. As stated above, the equilibrium toward pro-oxidation can be changed in the presence of transition metals such as Fe^3+^, Cu^2+^ and Mn ^2+^ to ^7+^ that facilitate the electron shuttle from flavonoid to molecular oxygen with the H_2_O_2_ generation in the process (Figure 2) [20,60].

Transition metals belong to a group of essential trace metals important for human nutrition and for that reason their levels in honey were often evaluated by multivariate analysis [74,75,76,77,78].

Most trace metals are bound to honey macromolecules, proteins, enzymes or are chelated by polyphenols. Regarding the latter, a differential distribution and quantitative differences in the content of the trace metals have been found between dark and light honeys that differed in the total polyphenolic content [78]. The first observation that darker honeys had a higher mineral content than light honeys was made by Anklam in 1998 [79]. Subsequent correlative studies revealed significantly higher quantities of transition metal ions Fe, Zn, Cu, and Mn in buckwheat, heather, and honeydew in comparison to light honeys such as acacia or rape honey [74,80,81,82].

A statistically significant correlation (*p* < 0.05) exists between the metal content and polyphenol content in honeys, indicating the polyphenols’ capacity in the chelation of these metals [79]. The high polyphenol and flavonoid content in dark honeys signifies the higher potential for complexation with transition metals. Stable chelate complexes result in an antioxidant effect but weak metal-polyphenol complexes can participate in the pro-oxidant activity with the generation of ROS (see Section 8).

## 7. Association between Transition Metals and Honey Color

Transition metals such as Fe, Cu, and Mn contribute to the honey color. They form colorful complexes when bonded with other molecules because, during this process, the unpaired electron of the d-subshell can be promoted to high energy d-orbital resulting in a color emission. The color of transition metals also depends on their oxidation state. Fe^3+^ in large concentrations expressed yellow-brown colors. However, Fe^3+^ complexation with polyphenols and its reduction to Fe^2+^ results in off-color development [32,83]. The yellow-brownish color can be restored by the addition of a strong chelator such as EDTA that blocks its re-binding to polyphenol.

Flavonoids and their quinones are also natural chromophores displaying a range of colors from light yellow to dark brown [83]. Moreover, the flavonoid oxidation to semiquinones and quinones generates yellowish colors while oxidatively polymerized phenolic acids and flavonoids increase browning [32]. 

With respect to honey, multivariate analyses, including CA, PCA or ANOVA, demonstrated a correlation between honey color and the mineral content. Using multiple linear regression (MLR), it has been shown that the color of the dark honeys (avocado, chestnut, honeydew, and heather) was correlated with the concentration of the trace elements such as arsenic, cadmium, iron, sulfur and lead and with calcium [52]. Chestnut honey (dark honey) possessed the highest content of Fe and Zn and the highest concentration of total polyphenol and antioxidant activity among other honey varieties (eucalyptus, citrus, multifloral and sulla). Significant correlations were found between metal content and polyphenol content in these honeys (*p* < 0.05) [78]. Bogdanov et al., 2007 [76] have shown that the concentration of trace metals, chromium, nickel, copper, zinc, manganese and iron, was the highest in dark honeys and decrease in the following order fir > chestnut > mountain blossom > rhododendron > mixed blossom > lime > rape > dandelion > acacia.

A strong relationship has been found between the content of manganese, copper and zinc and honey color coordinates’ (CIE L*a*b*) [74,81]. The principal component confirmed that dark buckwheat honey could be separated from other varieties due to strong correlations between its specific color parameters, the content of manganese, copper and zinc and antioxidant activity [81]. 

Thus, the color of honey is related to the content of transition metals, the content of polyphenols and their chelation capability (Figure 3).

## 8. Conditions Favoring the Switch from Antioxidant to Pro-Oxidant Activities

Due to redox cycling, the pro-oxidant and antioxidant activity of polyphenols could take place at the same time (Figure 2). Several factors play a role in changing the redox state of polyphenols; their concentrations, concentrations of transition metals, the ratio of metal to polyphenol in chelate complexes, the presence of other chelating compounds (such as vitamins, ascorbic acid, and tocopherol), and the pH of the solution. How these conditions might influence honey pro-oxidant activity?

### 8.1. Pro-Oxidation Depends on the Number and Position of Hydroxyl Group in Polyphenol Structure and Type of Transition Metal

The chelation of transition metals by polyphenols is recognized as an antioxidant activity that, by sequestering metal ions, prevents them from inducing the Fenton reaction and the generation of harmful hydroxyl radicals. At the same time, pro-oxidant activity may arise from the ability of polyphenols to reduce oxidized forms of the metal ion (Fe^3+^ or Cu^2+^) (Fe^2+^ or Cu^+^) in the chelate complex, generating H_2_O_2_ by superoxide radical dismutation and further catalyzing the Fenton reaction. Pro-oxidation results from the combined effect of polyphenol redox potential and the redox potential of Fe^3+^/Fe^2+^ couple or Cu^2+^/Cu^+^ couple [32]. Thus, the reduction of Fe^3+^ to Fe^2+^ (or Cu^2+^ to Cu^+^) in chelate complexes followed by metal auto-oxidation perpetuates redox cycling. This process is the main mechanism of pro-oxidant action and the formation of H_2_O_2_ [32,73].

The reduction potential of polyphenols depends on the number and position of metal-binding hydroxyl groups in their structure, the type of transition metal ions involved in the interactions with polyphenols and the stoichiometry (the ratio of metal to polyphenol molecules in chelate complexes). As shown in Figure 2A, there are three binding sites for transition metals in the flavonoid structure: (a) the 3′- and 4′-hydroxyl groups of the B ring, (b) the 3-hydroxyl and the 4-carbonyl groups in the C ring, and (c) the 5-hydroxyl group of the A ring [84,85]. The metal binding to a single 5-OH or 7-OH group in A ring provides weak or no activity while binding to catechol 3′-4′ OH groups of quercetin and pyrogallol 3′-4′-5′ OH of myricetin significantly increases both antioxidant and pro-oxidant activities [56,73]. In general, flavones and isoflavones commonly present in honey such as quercetin, kaempferol, fisetin, myricetin, morin, and rutin, daidzein, genistein, respectively, showed the Fe^3+^ reducing capacity in ferric-reducing ability power (FRAP) assays [86] in contrast to flavanones such as naringenin and hesperitin [73,85,87]. 

Myricetin and quercetin showed the highest Fe^3+^ to Fe^2+^ reduction potential and pro-oxidant effect when in high concentrations, with subsequent H_2_O_2_ production [73,83,84,87]. Firuzi et al. [88] provided an extensive list of the reduction potentials of polyphenols measured by the FRAP method.

Similarly, polyphenolic acids such as protocatechuic acid and chlorogenic acid showed a potent iron-reducing ability [89]. Gallic acid with three OH groups at ortho positions, and caffeic and chlorogenic acids with two OH groups in ortho position produced pro-oxidant activity resulting in DNA damage [56,90]. These phenolic acids are highly abundant in honey.

The possibility of a pro-oxidant effect increases when flavonoids coordinate Cu^2+^ ions instead of Fe^3+^ ions [32]. Most flavonoids have a higher reduction potential for copper than for iron to the difference in standard reduction potential of the Cu^2+^/Cu^+^ and Fe^3+^/Fe^2+^ (+0.15 V vs. +0.77 V, respectively) [87]. For example, myricetin and taxifolin efficiently reduced the Cu^2+^ to Cu^+^ causing the pro-oxidant effect and generation of hydroxy radicals from H_2_O_2_ via the Fenton reaction. Similarly, the binding of Cu^2+^ to the 3′–4′ site of quercetin also led to a rapid reduction of Cu^2+^ to Cu^+^ [85,91]. 

The reduction potential of flavonoids and the pro-oxidant effect decreases in the following order: Cu^+^ > Cu^2+^ > Fe^2+^ > Fe^3+^ [91,92,93].

### 8.2. Pro-Oxidation Effect Increases in the Presence of Strong Chelators 

Redox cycling plays a key role in the switch from antioxidant to prooxidant activity [32,73,94]. The presence in the redox reactions of other compounds (ligands) capable of chelating metals more efficiently than flavonoids can change flavonoid activity toward pro-oxidative activity. For example, EDTA, bleomycin or ADP are strong chelators often used as competitive ligands in experiments assessing the pro-oxidant properties of flavonoids [32]. EDTA forms Fe^2+^ or Fe^3+^ stable EDTA complexes in the presence of flavonoids, enhancing its auto-oxidation of free flavonoids [73,95]. Myricetin autooxidation increased eight-fold in the presence of Fe^3+^- EDTA complexes. However, in the presence of weaker ligands such as ascorbic acid, myricetin exhibited antioxidant activity, because it remained in complex with iron [73]. 

In comparison to EDTA, ascorbic acid is a weaker chelator of metal ions. However, its presence in the redox cycling reaction adds to the pro-oxidant activity of polyphenols in a dual way. Firstly, one-electron transfer from ascorbic acid to Fe^3+^ leads to its reduction to Fe^2+^ and subsequent participation in the Fenton reaction. Secondly, by donating one electron to Fe^3+^, ascorbic acid is oxidized to an ascorbate radical anion that can react with the flavonoid quinone to produce semiquinone radicals (Figure 2B), perpetuating redox cycling [96]. Thus, although flavonoids can act as pro-oxidants alone by reducing Fe^3+^ to Fe^2+^ or Cu^2+^ to Cu^+^ and initiating the Fenton reaction, the “help” of other reducing agents such as ascorbate radicals or superoxide radical anions (O_2_^•−^) that support the regeneration of quinone can promote redox cycling (Figure 2) [84]. 

### 8.3. Effect of pH

There is a general agreement that pro-oxidant activity occurs most efficiently under neutral or alkaline conditions [21,22]. A decrease in hydrogen ions at a pH above 7 facilitates the deprotonation of hydroxyl groups involved in metal binding, a prerequisite for both antioxidant and pro-oxidant effects (Figure 2A,B).

Much of our understanding of the metal-dependent pro-oxidant activity of polyphenols that results in H_2_O_2_ production has come from in vitro studies and from studies of H_2_O_2_ production in beverages (tea, coffee, beer, wine, and fruit juices) [21,96,97]. The average pH of tea and coffee is 5.5. Despite the acidic pH that inhibits the deprotonation of polyphenol OH metal-binding sites, catechin-rich tea and coffee rich in chlorogenic and caffeic acids, can produce up to 100 µM of H_2_O_2_ produced via polyphenol auto-oxidation [21]. The average pH of honey ranges between 3.5 and 5.5 which could be considered even more inhibitory for polyphenol autoxidation as compared to coffee and tea. However, several aspects of polyphenol autooxidation can compensate for the effect of low pH (see below), allowing the cautious conclusion that the non-enzymatic production of H_2_O_2_ in honey is possible. The important parameters influencing the pro-oxidant effect include the concentration and type of metal ions, the stoichiometry of the metal-to-polyphenol ratio, stability of the complex, the reduction potential of polyphenols, metal solubility in aqueous solutions and pH-dependent removal of Fe or Cu from chelate complexes. 

### 8.4. Stoichiometry

The pH-dependent deprotonation affects the stoichiometry in the metal–polyphenol complexes, their stability and their reduction potential. The stability of the Fe^3+^–flavonoid complex increased when Fe^3+^ ions coordinated more than one flavonoid ligand. The increased stability of the complex lowered the reduction potential of flavonoids and with it, the potential for auto-oxidation [73]. At pH 5, flavonoids with a higher number of hydroxyl groups such as taxifolin, myricetin and quercetin form complexes with Fe^2+^, or Fe^3+^ or Cu^2+^ with 1:2 and 2:1 stoichiometry (one metal ion coordinated by two ligands or two metal ions bound to two binding sites on one ligand) [98,99]. The number of ligands the metal is binding affects the stability of the metal-polyphenol complex and reduces the polyphenol reduction potential (see below). Despite the increased stability of the complex at 1:2 stoichiometry, myricetin and quercetin were still able to reduce Fe^3+^ and be auto-oxidized at pH 5. Binding to rutin, catechin and taxifolin at a stoichiometry of 1:2 only moderately lowered their reduction potential, while kaempferol and luteolin were poor reductants [32,87].

### 8.5. Solubility

While neutral and alkaline conditions reduced flavonoid auto-oxidation and the potential of the H_2_O_2_ generation, the availability (concentration) of iron ions for polyphenol binding is enhanced by its increased solubility in acidic pH. In an acidic aqueous solution, Fe^3+^ is in a soluble form. With increased pH, the complexes formed by Fe^3+^ coordination of multiple OH groups became insoluble and precipitate as dark flakes in an aqueous solution [87]. These results revealed that at an acidic pH, the ferric ions formed both soluble complexes and insoluble precipitates [83] with polyphenols, whereas Fe^2+^ polyphenols complexes were soluble. 

Moreover, low pH can cause the reductive disruption of Fe^2+^ and Fe^3+^–flavonoid complexes and the release of metal ions, initiating the Fenton reaction. Low pH also increases the susceptibility of the Fe^3+^ in chelate complexes to reduction [100]. A summary of factors influencing the pro-oxidant effect is schematically presented in Figure 4. 

### 8.6. Formation of Metal-Phenolic Networks

Flavonoids with multiple OH binding sites (myricetin, quercetin, and luteolin) can form Fe^3+^-coordinated metal-phenolic superstructures. It is assumed that the Fe^3+^ binding to pyrogallol or catechol groups of flavonoid forms coordinate complexes of mono-, bis-, or tri-stoichiometry. These complexes were able to self-assemble into an organized network structure of change functionality. For example, the metal–phenolic network adhered to colloidal particles and was found to cause liposome aggregation in the emulsion model evoking a pro-oxidant effect [99,101]. These effects are highly relevant to the formation of honey colloidal structures and H_2_O_2_ production [102]. Bijlsma et al., 2022, have found that the metal phenolic network was responsible for food discoloration [83]. 

The various studies on the pro-oxidant activity of metal–polyphenol complexes were conducted in in vitro model systems and may not represent conditions governing pro-oxidant activity in food such as honey. In a recent review on the antioxidant activity of polyphenols in oil in water (*o/w*) emulsion systems, it appeared that an aqueous medium influenced the partitioning of hydrophobic polyphenols into the emulsion interphase phase. Polyphenol structure and hydrophobicity determined the rate of their oxidation by transition metals that also co-partition with polyphenols at interphase. The antioxidant potential of phenolic compounds was found to decrease with increasing hydrophobicity. Quercetin exerted a pro-oxidant in this model. Caffeic acid was also auto-oxidized in the presence of Fe^3+^ at low pH 3 but became an antioxidant at pH 6 [103]. 

It could be concluded that the results on antioxidant/pro-oxidant activity of polyphenols from in vitro models do not always correspond to those obtained under conditions more like existing in food.

## 9. Association between Polyphenols Autooxidation, Formation of Higher-Order Structures and H_2_O_2_ Production 

The formation of quinones from polyphenol/flavonoid oxidation has far-reaching consequences for the structure and function of honey, including H_2_O_2_ production. The oxidation of polyphenols to semiquinone radicals and quinones results in fast reactions with a wide range of compounds including proteins, peptides, amino acids, di-carbonyl compounds (such as glyoxal and methylglyoxal), carbohydrates, lipid oxidation products, intermediates of Maillard reactions, and compounds formed during thermal degradation of carbohydrates [104,105,106]. Moreover, semiquinone radicals participate in radical–radical binding generating oligomers enriched in hydroxyl groups that may either enhance their radical scavenging activity or increased their pro-oxidant activity when complexed with proteins [107,108,109].

Most of our knowledge about the pro-oxidant activities of polyphenols presented above comes from in vitro model systems. However, in nature, hydrophobic polyphenols rarely exist in free forms as aglycones in hydrophilic solutions such as honey. Rather, they are conjugated with hydrophilic molecules such as proteins or/and carbohydrates. These structural modifications may enhance or decrease their antioxidant/pro-oxidant activities. In honey, both quinones and polyphenols formed complexes with proteins that self-assemble into higher-order structures [24,28,102]. Two types of high molecular weight complexes isolated from honey differed in molecular size, the ratio of polyphenol to protein and antioxidant/pro-oxidant activity; the high molecular weight (180–230 kDa) complexes contained unoxidized polyphenols with the high protein to polyphenol ratio, and carried antioxidant activity, whereas lower molecular weight complexes (85–110 kDa) were instead composed of quinone–protein conjugates where quinones were a dominant partner and carried pro-oxidant activity [28,110]. Since natural buckwheat honey contained increased amounts of iron and copper ions [75,82], this pro-oxidant effect was possibly due to the quinones’ ability to chelate Fe^3+^ and Cu^2+^ and to reduce them.

### 9.1. Association between Melanoidin Formation and H_2_O_2_ Production

Quinone reactivity toward proteins and the Maillard reaction products appeared to be a primary step in non-enzymatic browning and the formation of melanoidins in honey. Principal component analysis revealed that the MRPs (Maillard Reaction Products) significantly contributed to honey antioxidant activity measured by ORAC (r = 0.938, *p* < 0.0001). In addition, the extremely significant correlations between ORAC, MRPs, phenolic content and honey color suggested that both MRPs and phenolic compounds exert their antioxidant activity while being part of a higher molecular weight structure [25]. In contrast to coffee melanoidins, honey melanoidins are built on a polyphenol–protein backbone with the participation of oligosaccharides, as indicated by LC-ESI-MS [33]. Thus, both MRP and polyphenols in melanoidins could exert pro-oxidant effects and contribute to H_2_O_2_ production because melanoidins are known metal chelators [111]. Moreover, melanoidins added to the brown color of honey. 

### 9.2. Association between the Formation of Colloidal Particles and H_2_O_2_ Production

Accumulated evidence from our laboratory indicates the spontaneous formation of higher-order structures in honey. We currently do not know the chemical nature of these structures but tend to speculate that they comprise both the aggregates of polyphenol-protein complexes and/or melanoidins that self-assemble to large colloidal particles in viscose of honey solution [24,103]. An important feature of honey colloids is their aggregation and disintegration upon honey dilution. Based on the DLS (Dynamic Light Scattering) and polydispersity data, the particle self-assembly and disintegration during honey dilution attained the inverted U-shaped curve that paralleled the inverted U-shaped- curve of H_2_O_2_ production and degradation in the same dilution range [24]. The formation and dissociation of colloidal particles undergo a phase transition from a colloidal structure to dispersion at a certain point of dilution. The phase-transition point coincided with the maximum peak of H_2_O_2_ production in dark and medium honeys suggesting that the formation of particles was associated with the production and accumulation of H_2_O_2_ while the degradation of particles was associated with the cessation of the H_2_O_2_ production [24,102]. 

This novel finding on the coordination between the production of H_2_O_2_ and the aggregation–dissociation states of honey colloidal structure sets the stage for future efforts to discern the mechanism that leads to biphasic production and accumulation of H_2_O_2_.

## 10. Conclusions

Based on the results of several multivariate analyses, and our own experimental data, the study led to defining the combination of different compounds which allows more accurate prediction of non-enzymatic H_2_O_2_ production. Honey color appeared as a parameter that integrated polyphenol-dependent pro-oxidant reactions resulting in H_2_O_2_ generation (Figure 5).

The intensity of honey color was strongly correlated with the content of polyphenolic acids, flavonoids, and transitional metals (*p* < 0.001). The color intensity was signaling the quantitative differences in the content of compounds required for pro-oxidant activity such as polyphenols and transition metals and the capability of polyphenols containing catechol or pyrogallol structures for redox cycling and the H_2_O_2_ production. In general, dark color honeys predominantly contained higher contents of color-impending constituents (flavonoids, their quinones, melanoidins) together with higher antioxidant/pro-oxidant activities in the presence of elevated content of transition metals, Fe and Cu. Experimental and correlative data indicated that the H_2_O_2_ production in dark honeys was higher than in medium and light honeys and could be explained by higher concentrations of transition metals. Figure 5 is a quintessence of these relationships.

The polyphenol redox reactions and formation of highly reactive quinones could further contribute to the non-enzymatic production of H_2_O_2_ by the formation of the higher order structure such as melanoidins and colloidal particles. Due to the incorporation of polyphenols and MRPs into melanoidins and colloidal particle structures, these complexes are endowed with metal-chelating and reducing capacity, participating in a pro-oxidant effect and H_2_O_2_ production, such as that observed by melanoidins of coffee and tea. 

The additional, non-enzymatic formation of H_2_O_2_ and the production of hydroxyl radicals via the Fenton reaction could significantly increase the antibacterial potency of honey, creating an exciting possibility from a therapeutic point of view. Honey color can be used as a guiding principle since it connects all necessary compounds involved in the non-enzymatic production of H_2_O_2_. 

## Figures and Tables

**Figure 1 metabolites-13-00526-f001:**
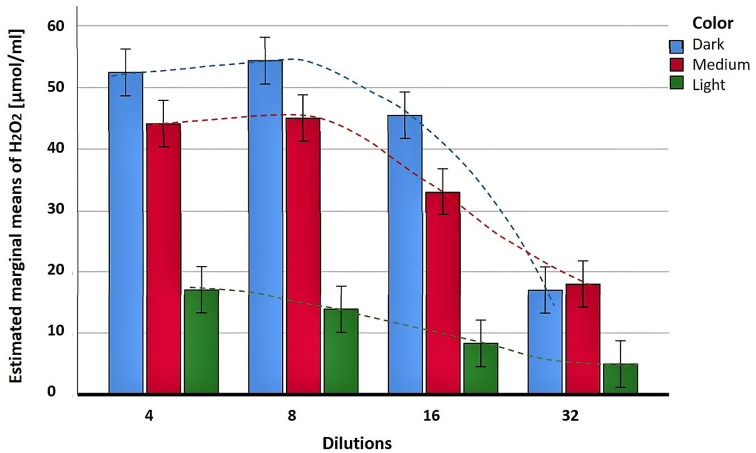
The content of H_2_O_2_ produced in dark-, medium- and light- color in honeys (columns). The non-linear relationship between the production of H_2_O_2_ and honey dilution is presented as the inverted U- shaped curve (dotted lines).

**Figure 2 metabolites-13-00526-f002:**
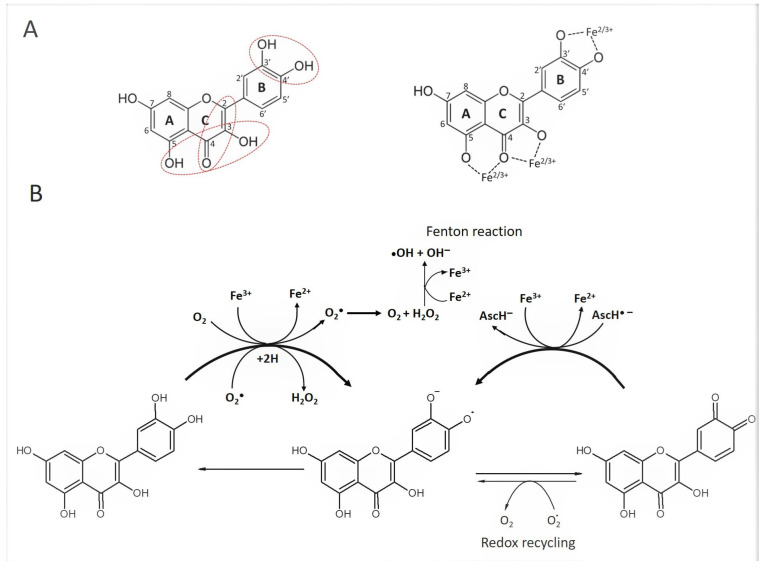
The potential mechanism of non-enzymatic H_2_O_2_ production in honey by flavonoid auto-oxidation. (**A**) Structural elements required for an efficient radical scavenging activity and metal ion-chelation are circled [58,60]. Three possible binding sites of a metal cation to a flavonoid. (**B**) A schematic proposal of H_2_O_2_ production in honey via flavonoid (quercetin) autooxidation based on literature data (detailed description in the text). Redox cycling and pro-oxidant activity of flavonoid is promoted in the presence of ascorbic acid. Ascorbic acid (AscH^−^) is oxidized to ascorbate radical anion (Asc^•−^) by one-electron transfer to Fe^3+^ and its reduction to Fe^2+^. Ascorbate radical (Asc^•−^) reduces quinone to semiquinone radicals propagating redox cycling.

**Figure 3 metabolites-13-00526-f003:**
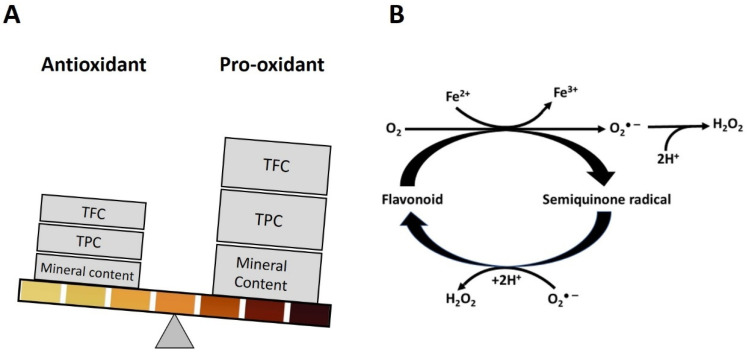
A schematic depiction of the quantitative differences in the total polyphenolic content (TPC), total flavonoid content (TFC) and mineral content in the relation to honey color (based on correlative coefficients of Table 2) that change the antioxidant- pro-oxidant balance (**A**) and produce a marked flux of H_2_O_2_ (**B**). The gradient of honey color illustrates the corresponding increase of TFC, TPC and mineral content.

**Figure 4 metabolites-13-00526-f004:**
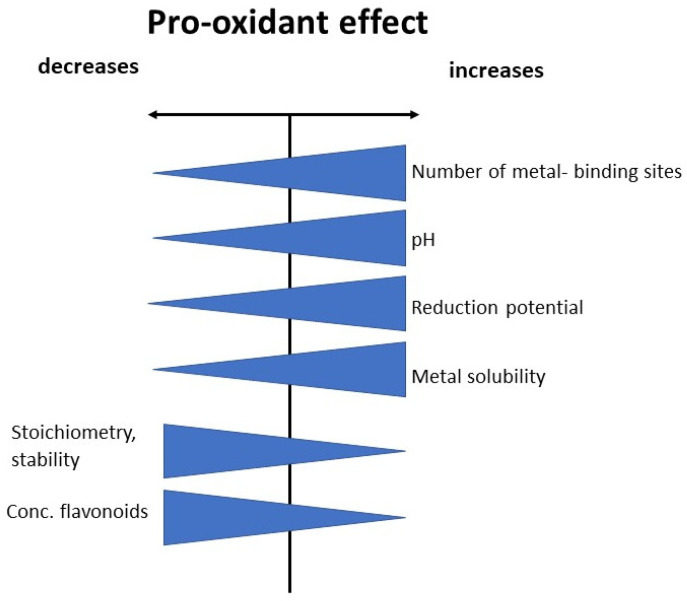
Summary of factors influencing pro-oxidant effect.

**Figure 5 metabolites-13-00526-f005:**
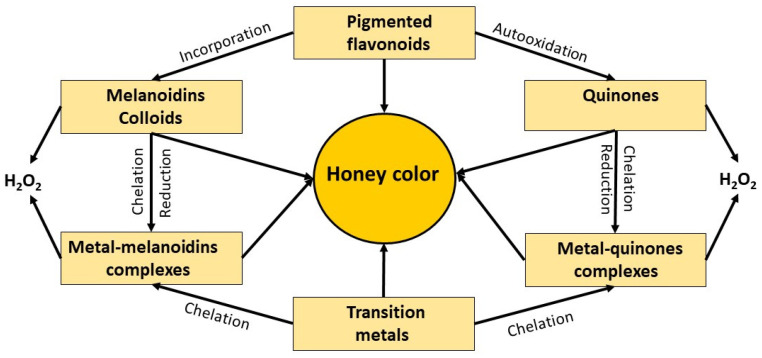
A diagram representing honey color as the direct result of the combine effect of the content of the color-impending compounds (flavonoids, quinones, transition metals) and the color-developing reactions (polyphenol autooxidation, quinone-protein conjugation, metal- quinone complexes, Maillard reactions and the formation of melanoidins and colloidal particles) and its indirect association with H_2_O_2_ production.

**Table 1 metabolites-13-00526-t001:** List of honeys used and their characteristics.

Honey Sample.	Plant Source	Color * (A560–720 nm)	Total Phenolic * (ug GAE/g of Honey)	ORAC * (uM TE/g of Honey)	H_2_O_2_ Production ** (Fluorescence)
Dark					
H77	Buckwheat	1.27	513.25	19.13	1319.3
H76	Buckwheat	0.72	318.74	19.77	1208
H226	Buckwheat	0.62	362.86	15.45	1152
H23	Buckwheat	1.03	410	15.28	1210
H149	Buckwheat	0.88	408.03	12.75	1470
Medium					
H221	Buckwheat (light)	0.38	144	5.31	886
H208	Buckwheat	0.37	143.55	4.86	680
H11	Wildflower/clover	0.13	106.97	5.37	1280
Light					
H210	Wildflower	0.3	120.2	4.74	560
H20	Sweet clover/buckwheat	0.24	83.75	4.49	1220
114	Sunflower	0.12	80.82	2.75	424
H62	Borage	0.05	66.45	2.84	529

* Methods used are described in details in [24], ** H_2_O_2_ measurements were conducted using Amplex Red method, as described [6].

**Table 2 metabolites-13-00526-t002:** Correlation matrix (all correlations are significant, *p* < 0.05).

		Honey Color	TPC	TFC	MRP	Antioxidant			
						ABTS	FRAP	DPPH	ORAC
Honey color	Average	1.00	0.89	0.93	0.98		0.90	0.86	0.83
	Range		0.68–0.99				0.89–0.92	0.84–0.88	0.73–0.93
TPC	Average	0.88	1.00	0.86	0.95	0.97	0.89	0.86	
	Range	0.68–0.99		0.71–0.98		0.72–0.96	0.87–0.92		
TFC	Average	0.87	0.97	1.00					
	Range	0.77–0.98	0.73–0.98						
MRP	Average	0.98	0.95		1.00			0.94	
	Range								
Anti- oxidant: ABTS	Average		0.85			1.00			
	Range		0.72–0.96						
Anti- oxidant: FRAP	Average	0.92	0.91			0.80	1.00		
	Range		0.95–0.88			0.72–0.89			
Anti- oxidant: PPH	Average	0.88	0.90				0.89	1.00	
	Range		0.89–0.92						
Anti- oxidant: ORAC	Average	0.83	0.86		0.94		0.82		1.00
	Range	0.93–0.73	0.86–0.87						

**Table 3 metabolites-13-00526-t003:** Pearson correlation coefficients and correlation matrix.

Variable	Color	Total Phenolic	ORAC	H_2_O_2_ Production
Color	1.000	0.9721 *p* < 0.0001	0.8892 *p* < 0.0001	0.6512 *p* = 0.0218
Total phenolic	0.9721 *p* < 0.0001	1.000	0.9259 *p* < 0.0001	0.6826 *p* = 0.0144
ORAC	0.8892 *p* < 0.0001	0.9259 *p* < 0.0001	1.000	0.6512
H_2_O_2_ production	0.6512 *p* = 0.0218	0.6826 *p* = 0.0144	0.6823 *p* = 0.0145	1.000

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
