# Peer review of "Unexpected Value of Honey Color for Prediction of a Non-Enzymatic H2O2 Production and Honey Antibacterial Activity: A Perspective"

_metabolites, 2023, doi:10.3390/metabo13040526_

Round 1
Reviewer 1 Report
Intro can be improved Observe the general formatting of the text as some examples: H2O2 (H2O2), p<0.001 in the table is P<0.001, Fe+3 (Fe+3). Observe and correct spaces between letters throughout the text Table 1 is in the text, above the place where it is cited, it should be changed places. In the text table 2 appears before table 1. On line 255 - correct dyhydroxyflavone font formatting and size When citing the author in the text, check the need to include the year. Example on line 373 has author without year and elsewhere has author and year. On line 515, 518-522 and 526 - underlined words. In line 538 the word polyphenol is wrongly separated check The conclusion must be rethought, it is too long, with a figure, the ideal conclusion is that it be shorter, to close what was observed in the work. If the information currently in the conclusion is important, check how to add it in the text, outside the conclusion. A comparison could also have been made with Manuka honey, which is very well known and there are several works that discuss the H2O2 of this honey. And also written about monofloral honeys, discussing more about the color, phenolics and antioxidant action. As it is a review article, it is necessary to improve the discussion throughout the work, go deeper, there are recent works published in the journal Antibiotics, a special edition Antioxidant and Antibacterial Properties of Honey, several works there can help.
Author Response
Intro can be improved
The Introduction now contains additional sentence emphasizing the significance of H2O2 for antibacterial activity of honey:
Line 42-48: Although honey possesses various secondary metabolites with antimicrobial activities originating from plant (flavonoids, phenolic acids, volatile compounds, pathogenesis-related proteins), microbes (antimicrobial peptides, biosurfactants, siderophores) and honeybees (defensins, hymenoptaecins, jelleins) [13], none of these compounds alone can kill or supress growth of a broad spectrum of microorganisms to the degree comparable with H2O2. Removal of H2O2 by pre-treatment of honey with catalase reduces honey antibacterial activity by 50-80%,
Observe the general formatting of the text as some examples: H2O2 (H2O2), p<0.001 in the table is P<0.001, Fe+3 (Fe+3). Observe and correct spaces between letters throughout the text
The formatting has been corrected through out the manuscript, including subscripts, and changes to lower case p in Tables.
Table 1 is in the text, above the place where it is cited, it should be changed places. In the text table 2 appears before table 1.
The Table 1 has now been denoted Table 2, and is placed in the correct place (section 3.2) while original Table 2 is now Table 1 and precedes the Table 2.
On line 255 - correct dyhydroxyflavone font formatting and size
Regarding the font issues: I noticed that my Windows had randomly changed the font, style or introduced underlines on its own. In my latest version of the manuscript, there are no changes of fonts or underlined words. I checked it.
When citing the author in the text, check the need to include the year. Example on line 373 has author without year and elsewhere has author and year.
Corrected
On line 515, 518-522 and 526 - underlined words.
Please, see above explanation regarding random changes of font styles by my computer. The problem has been fixed.
In line 538 the word polyphenol is wrongly separated check
Line 538: I cannot find this word in my text.
The conclusion must be rethought, it is too long, with a figure, the ideal conclusion is that it be shorter, to close what was observed in the work. If the information currently in the conclusion is important, check how to add it in the text, outside the conclusion.
The Conclusions were drastically shortened as the Reviewer suggested. I also introduced Fig. 4 as a graphical abstract.
A comparison could also have been made with Manuka honey, which is very well known and there are several works that discuss the H2O2 of this honey.
Hydrogen peroxide is absent or in insignificant amounts in Manuka honey and for that reason Manuka honey is not included in this review. Instead of H2O2, methylglyoxal is associated with antibacterial activity of this honey and is termed “non-peroxide activity”. As such, it is beyond scope of this review. I included in the Introduction the explanation, why Manuka honey is not part of this review.
Line 57-59: This review is based on honey of the European and North American origin and did not include Manuka honeys of New Zealand and Australia, derived from Leptospermum scopiarum, which are apparently devoid of H2O2.
And also written about monofloral honeys, discussing more about the color, phenolics and antioxidant action. As it is a review article, it is necessary to improve the discussion throughout the work, go deeper, there are recent works published in the journal Antibiotics, a special edition Antioxidant and Antibacterial Properties of Honey, several works there can help.
This review is sharply focused on non-enzymatic synthesis of H2O2 via polyphenol autooxidation in both groups of honeys, monofloral and multifloral. I examined data from several multivariate analyses which included both monofloral and multifloral honeys. I am very familiar with the Special Issue of the Antibiotics, dedicated to honey. I also published there my work.
Reviewer 2 Report
Brudzynski has nicely compiled different scientific reports for this comprehensive review.
There are some minor comments for further improvement. The tables in the review are not properly aligned. Please also check the font style and size. It is sometimes not consistent. Some text was also underlined in section 9.2 which is unnecessary.
Please give the full forms of the abbreviations like DPPH, ABTS when you first mention them.
The last line of section 3.1 should be rephrased and in the same line it should be interdependent.
In section 3.2, Paragraph 2 - the sentence "using the data of table 1...." should be rephrased as the grammar is not correct. and the table where this data is shown is table 3 (in the text it says table 1).
In section 8.1, last paragraph - the sentence "the majority of flavonoids....." is not clear and should be rephrased.
The last line in section 8.2 " the beyond involve in..." is not correct english and is not clear. Please rephrase it.
In section 8.4, "Despite that, myricetin...." should be rephrased so that its more clear.
There are some grammatical errors in the text that should be corrected.
Author Response
There are some minor comments for further improvement. The tables in the review are not properly aligned. Please also check the font style and size. It is sometimes not consistent.
The Tables are now properly aligned.
The font and style are now verified. Some text was also underlined in section 9.2 which is unnecessary.
I noticed that my Windows had randomly changed the font, style or introduced underlines on its own. In my version of the manuscript, the font size and style is uniform (Times Roman, 12) and there is no underlined words. I set up default setup and pressed the Font Button for entire length of the manuscript. I hope, it works!
Please give the full forms of the abbreviations like DPPH, ABTS when you first mention them.
Line 157-159: (measured by Folin- Ciocalteau method, FC) and flavonoid content (measured by aluminum chloride method) and antioxidant activity (measured by DPPH (2,2-diphenyl-1-picryl-hydrazyl-hydrate), FRAP (Ferric Reducing Antioxidant Power) or ORAC (Oxygen radical absorbance capacity assay),
The last line of section 3.1 should be rephrased and in the same line it should be interdependent.
Corrected
In section 3.2, Paragraph 2 - the sentence "using the data of table 1...." should be rephrased as the grammar is not correct. and the table where this data is shown is table 3 (in the text it says table 1).
Yes, its now corrected. Thank you.
In section 8.1, last paragraph - the sentence "the majority of flavonoids....." is not clear and should be rephrased.
Line 405: Most flavonoids…
The last line in section 8.2 " the beyond involve in..." is not correct english and is not clear. Please rephrase it.
Line 450-453: The important parameters influencing pro-oxidant effect include: concentration and type of metal ions, stoichiometry of metal to polyphenol ratio, stability of the complex, the reduction potential of polyphenols, metal solubility in aqueous solutions and pH-dependent removal of Fe or Cu from chelate complexes.
In section 8.4, "Despite that, myricetin...." should be rephrased so that its more clear.
Line 464-465: Despite increased stability of the complex at 1:2 stoichiometry, myricetin and quercetin were still able to reduced Fe3+ and auto-oxidised at pH 5.
There are some grammatical errors in the text that should be corrected.
I re-checked and corrected the manuscript for grammatical errors.
Reviewer 3 Report
The article overall is well written and the topic is within the interest, but the article needs carefull revision. I suggest some points and comments and recommend for minor revision.
1. Kindly put attention to the fonts in the article. The font should be uniform throughout the article. Line 129, 515, 518-522,there are different fonts used. Kindly correct it and carefully revise the whole paper.
2. Line 281, 371, 381-388, 2+ for the metals should be written in superscript.Cu2+
3. Line 455, “Biding site” I think it should be “binding site.
It should be great if the author can add some more interesting diagrams, as diagram are considered as attractive point and backbone of the article.
Author Response
- Kindly put attention to the fonts in the article. The font should be uniform throughout the article. Line 129, 515, 518-522, there are different fonts used. Kindly correct it and carefully revise the whole paper.
I noticed that my Windows had randomly changed the font, style or introduced underlines on its own. In my version of the manuscript, the font size and style were uniform (Times Roman, 12) and there were no underlined words. I set up default setup and pressed the Font Button for entire length of the manuscript. Then, I check the whole text to be sure that the font size and style are correct.
- Line 281, 371, 381-388, 2+ for the metals should be written in superscript.Cu2+
The metal ions oxidation state is now uniformly denotated as superscripts through out the manuscript.
- Line 455, “Biding site” I think it should be “binding site.
Corrected
It should be great if the author can add some more interesting diagrams, as diagram are considered as attractive point and backbone of the article.
The diagram has been designed and introduced into the text as a Figure 4 (page 21).
Reviewer 4 Report
The present review manuscript entitled "Unexpected value of honey color for prediction of a non-enzymatic H2O2 production and honey antibacterial activity. A perspective" by Katrina Brudzynski involved the bibliographical analyses of current data regarding the colour differences found in a diversity of honey varieties as an indicator of their composition.
The introduction is complete and according to the developed topic of the manuscript, and it has updated bibliographical references to support the research.
Also, the manuscript is interesting, clear, organize, and focused on the topic that is of growing interest due to the current interest of honey bee products and their potential sources of biomarkers to analyse their biological activities.
Moreover, the information they described is supported with clear and logical figure that summarize all the required data for the discussion item. According to this point it should be appropriate to check the quality of Figure 2 (the molecules look blurry).
Besides, it should be important to show which is the “dark”, “medium”, and “light” colour in honeys in Figure 1 to specify this information.
Furthermore, some typo mistakes should be corrected (in yellow, pdf file attached) such as:
· There are some blank spaces between words throughout the manuscript.
· H2O2 as a molecular formula needs to use the subscript for the atomic number: H2O2. Please check this mistake throughout the manuscript.
· Line 33: Hydrogen peroxide is the main antibacterial compounds of honey. The H2O2 is the result of several reactions from the honey but it could be confused to consider it as metabolite (compound from the honey).
· Please check the correct nomenclature for the metallic ions and the radicals (subscripts and superscripts should be use it)
· Topic 9.2. Is it necessary to underline the paragraph? As a suggestion it should be more appropriate to said “we currently don’t have the information of the chemical structures of these compounds.
· As a suggestion: please use one final table in section 8 to condense all the information of each subsection (8.1 to 8.6). This improvement will increase the reader’s attention.
· Figure 4: according to the review analyses it should be interesting to show the colour ranges for each part of the diagram.
Finally, I would like to invite the authors to include the abbreviation list of words at the end of this manuscript.

Author Response
The present review manuscript entitled "Unexpected value of honey color for prediction of a non-enzymatic H2O2 production and honey antibacterial activity. A perspective" by Katrina Brudzynski involved the bibliographical analyses of current data regarding the colour differences found in a diversity of honey varieties as an indicator of their composition.
The introduction is complete and according to the developed topic of the manuscript, and it has updated bibliographical references to support the research.
Also, the manuscript is interesting, clear, organize, and focused on the topic that is of growing interest due to the current interest of honey bee products and their potential sources of biomarkers to analyse their biological activities.
Moreover, the information they described is supported with clear and logical figure that summarize all the required data for the discussion item. According to this point it should be appropriate to check the quality of Figure 2 (the molecules look blurry).
The resolution of Figure 2 was increased to over 1000 pixels.
Besides, it should be important to show which is the “dark”, “medium”, and “light” colour in honeys in Figure 1 to specify this information.
The color of columns is specified at the top, right side of the Figure 1
Furthermore, some typo mistakes should be corrected (in yellow, pdf file attached) such as:
- There are some blank spaces between words throughoutthe manuscript.
The blank spaces are now filled.
Line 450-453: The important parameters influencing pro-oxidant effect include: concentration and type of metal ions, stoichiometry of metal to polyphenol ratio, stability of the complex, the reduction potential of polyphenols, metal solubility in aqueous solutions and pH-dependent removal of Fe or Cu from chelate complexes.
- H2O2 as a molecular formula needs to use the subscript for the atomic number: H2O2. Please check this mistake throughout the manuscript.
I corrected all denotation of H2O2 in the manuscript
- Line 33: Hydrogen peroxide is the main antibacterial compounds of honey. The H2O2is the result of several reactions from the honey but it could be confused to consider it as metabolite (compound from the honey).
It is now: “H2O2 is produced as a by-product of glucose oxidation by honey bee enzyme, glucose oxidase,….
- Please check the correct nomenclature for the metallic ions and the radicals (subscripts and superscripts should be use it)
All metal ions oxidation states are denoted using subscripts
- Topic 9.2. Is it necessary to underline the paragraph? As a suggestion it should be more appropriate to said “we currently don’t have the information of the chemical structures of these compounds.
I noticed that all Reviewers complained about formatting issues (font change, size of the font, underlying). For unknown reason, my Windows had randomly changed the font, style or introduced underlines on its own. I used the Font Button to normalize the text and checked uniformity of font sizes and style.
- As a suggestion: please use one final table in section 8 to condense all the information of each subsection (8.1 to 8.6). This improvement will increase the reader’s attention.
Very good suggestion. I designed and introduced to the manuscript a new diagram summarizing the effects of different factors on pro-oxidant effect (Figure 4, page 21).
- Figure 4: according to the review analyses it should be interesting to show the colour ranges for each part of the diagram.
Most of the colors developed via redox reactions ranged from light yellow to brown. The same relates to different flavonoids. What is an actual color of mixture of flavonoids or their quinones is hard to predict. That is why I have chosen yellow as arbitrary color in this Figure.
Finally, I would like to invite the authors to include the abbreviation list of words at the end of this manuscript.
Abbreviation such as DPPH, ABTS, FC etc, are now included in the text at their first mention.
Round 2
Reviewer 1 Report
Foram feitas correcções.
Se for possível na edição melhorar a formatação dos espaços na tabela 1, seria conveniente.
Author Response
done